# The Circulating Selenium Concentration Is Positively Related to the Lipid Accumulation Product: A Population-Based Cross-Sectional Study

**DOI:** 10.3390/nu16070933

**Published:** 2024-03-23

**Authors:** Kunsheng Zhao, Yun Zhang, Wenhai Sui

**Affiliations:** National Key Laboratory for Innovation and Transformation of Luobing Theory, The Key Laboratory of Cardiovascular Remodeling and Function Research, Chinese Ministry of Education, Chinese National Health Commission and Chinese Academy of Medical Sciences; Department of Cardiology, Qilu Hospital of Shandong University, Jinan 250012, China; kunshengzhao@163.com

**Keywords:** selenium, lipid accumulation product, obesity

## Abstract

The lipid accumulation product (LAP) is a reliable marker of metabolic syndrome, which includes conditions like obesity. However, the correlation between the circulating selenium (CSe) concentration and the LAP is currently unclear. This study aimed to ascertain this correlation. Overall, 12,815 adults aged ≥20 years were enrolled in this study. After adjusting for all the confounding variables, CSe was positively correlated to the LAP (β = 0.41; 95% confidence interval [CI]: 0.28, 0.54; *p* < 0.001). Compared with the lowest quartile of CSe, the highest quartile of CSe was positively related to the LAP (β = 0.16; 95% CI: 0.12, 0.21; *p* < 0.001). Moreover, the correlation between CSe and the LAP revealed a positive non-linear trend. In the subgroup analysis, interaction effects were observed for age, sex, smoking, and stroke (*p* for interaction < 0.05). The effects were stronger for males (β = 0.64, 95% CI: 0.47, 0.80; *p* < 0.001) and individuals who smoke at the time of the trial (β = 0.64, 95% CI: 0.37, 0.91; *p* < 0.001). In conclusion, our results indicated that CSe was positively correlated with the LAP in a non-linear manner. Future research is warranted to explore their relationship and better understand the mechanisms underlying this association.

## 1. Introduction

Obesity is a global problem [1,2], contributing to the incidence and mortality of a range of diseases, including hypertension, coronary heart disease, type 2 diabetes mellitus (T2DM), dyslipidemia, cerebrovascular accidents, and cancer [3,4,5,6]. It also results in a substantial increase in healthcare costs. However, to date, no country has implemented a successful public health model to reduce the prevalence of obesity, despite continued efforts to do so [7]. In light of this, tackling obesity has become a global health priority.

Obesity is characterized by the growth of adipose cells and the enlargement of adipose tissue [8]. Body mass index (BMI) is a conventional, simple measurement that is applied to assess relative weight. However, BMI is neither specific for adipose or lean tissues, nor is it a reliable predictor of cardiovascular events or mortality [9,10]. The lipid accumulation product (LAP) is calculated from the waist circumference (WC) and triglyceride (TG), and is a reliable marker of metabolic syndrome, which encompasses obesity [11]. Furthermore, the LAP is superior to BMI in distinguishing cardiovascular risk and T2DM [11,12].

Selenium (Se), an essential trace element and constituent of selenoproteins, plays many vital physiological roles, including antioxidation, anti-inflammation, anti-aging, energy metabolism, immune regulation, redox signaling, cellular differentiation, protein folding, and gene expression [13,14,15,16,17,18,19,20,21,22]. Oxidative stress (OS), inflammation, endocrine function dysfunction, and energy metabolism disorders are key in the pathogenesis of obesity [8,23,24,25]. Thus, it is hypothesized that Se may influence obesity levels. Notably, a previous meta-analysis, including 65 articles, highlighted that the relationship between Se and being overweight or obese was controversial [26]. Furthermore, to date, the association between the circulating selenium (CSe) concentration and LAP has not been explored. Hence, based on the National Health and Nutrition Examination Survey (NHANES), this research aimed to assess the correlation between CSe and the LAP.

## 2. Materials and Methods

### 2.1. Study Population

The NHANES was conducted in order to make a health evaluation for all Americans [27]. The NHANES project was conducted according to the guidelines of the Declaration of Helsinki, and approved by the Institutional Review Board of the National Center for Health Statistics (NCHS) (protocol code: 2011-17, date of approval 2011; protocol code: 2018-01, date of approval 2018). A written notice was submitted to all adult individuals. The survey utilized open-access data from the NHANES. A secondary analysis in our study was based on the NHANES. The data analyzed in our study combined four survey cycles from the NHANES (2011–2012, 2013–2014, 2015–2016, and 2017–2018). After screening the data of 39,156 participants, 26,341 participants were eliminated due to the following reasons: Age < 20 years (*n* = 16,539), cancer (*n* = 2184), pregnancy (*n* = 245), missing CSe (*n* = 6423), missing WC (*n* = 684), and missing TG (*n* = 266). A total of 12,815 participants were eligible for inclusion and therefore incorporated into the analyses (Figure 1).

### 2.2. Acquisition of Variables

Age, sex, race, education, family poverty ratio of income (FPRI), marital status, and pregnancy status were acquired from the demographic data. The BMI, WC, and blood pressure were retrieved from the examination data. A history of various self-reported diseases, medication use status, alcohol consumption, and smoking status were acquired from the questionnaire data. Serum lipid, fasting plasma glucose, hemoglobin A1c (HbA1c), and whole blood Se levels were retrieved from laboratory data. More detailed information is available from the NHANES.

### 2.3. Case Definition

Hypertension was defined using the following criteria: Self-reported hypertension; diagnosed by a physician; patients taking antihypertensive medication; an average systolic blood pressure (SBP) of ≥130 mmHg; and/or diastolic blood pressure (DBP) of ≥80 mmHg [28]. T2DM was defined as follows: a diagnosis of diabetes mellitus; taking hypoglycemic medications or using insulin; HbA1c ≥ 6.5%; fasting plasma glucose ≥ 7.0 mmol/L; and/or a 2 h plasma glucose ≥ 11.1 mmol/L [29]. Stroke or coronary heart disease was defined as a self-reported stroke, or a coronary heart disease as diagnosed by a physician.

### 2.4. Lipid Accumulation Product Index Calculation

The LAP index was calculated from the WC and TG using the following formula [11]:LAP (males) = (WC [cm] − 65) × TG (mmol/L)
LAP (females) = (WC [cm] − 58) × TG (mmol/L)

### 2.5. Covariates Assessment

Covariates included age (≤44 [young], 45–59 [middle-aged], ≥60 years [old]) [30], sex (males, females), race (Mexican American, other Hispanic, non-Hispanic White, non-Hispanic Black, others), FPRI (<1, 1–3, >3, unavailable [Un]), education (<high school, high school, >high school, Un), BMI (<25, 25–30, ≥30 kg/m^2^, Un) [31], alcohol consumption (≤3, >3 drinks/day, Un), total cholesterol (TC) (<6.22, ≥6.22 mmol/L, Un) [32], TG (<2.26, ≥2.26 mmol/L, Un) [32], glucose (<6.1, 6.1–7.0, ≥7.0 mmol/L, Un) [29], diastolic blood pressure (DBP) (<80, ≥80 mmHg, Un) [28], systolic blood pressure (SBP) (<130, ≥130 mmHg, Un) [28], marital status (married, never married, others, Un), hypertension, T2DM (no, yes), stroke, current smoking status, coronary heart disease, current taking of hypotensive drugs, and current injection of insulin (no, yes, Un).

### 2.6. Statistical Analysis

Regarding the NHANES guidelines, the statistical analysis adopted suitable sampling weights. Categorical variables are presented as numbers (%), whereas continuous variables are presented as medians (interquartile range) for skewed distributions. Differences were calculated via applying the chi-square test for categorical variables and the rank sum test for continuous variables. The potential confounders were explored using univariate linear regression analysis. After adjusting for multiple factors in different models, multivariate linear regression analyses were used to analyze the independent correlation between CSe and the LAP. Covariates with *p* < 0.05 in univariate analysis were included in the multivariate analysis as adjusting confounders. Three different models were adopted to verify independent correlations according to the guidelines of the STROBE statement. Model A was not adjusted for any variables. Model B was adjusted for age, sex, and race. Model C was adjusted for age, sex, race, education, marital status, alcohol consumption, BMI, hypertension, T2DM, stroke, coronary heart disease, TC, glucose, SBP, DBP, current injection of insulin, and current taking of hypotensive drugs. Dose–response analysis was adopted to test linear or non-linear relationships after adjusting for the same variables in model C. Subgroup analyses were applied to explore the variable interactions. The natural logarithmic (Ln) transformation of CSe and the LAP was carried out due to its non-normal distribution.

All statistical analyses were performed using R (version 4.2.0) and EmpowerStats (version 5.0). *p* < 0.05 (two-sided) was deemed statistically significant.

## 3. Results

### 3.1. Baseline Characteristics

The participants were divided into three groups based on the LAP tertiles (Table 1). When compared with individuals in the low LAP, subjects in the high LAP were significantly more likely to be elderly, male, Mexican American, less educated, poor, married, and obese. They were also more likely to be currently injecting insulin or taking hypotensive drugs, or display symptoms of hypertension, T2DM, stroke, coronary heart disease, and higher levels of SBP, DBP, TC, glucose, and CSe (all *p* < 0.05).

### 3.2. Univariate Analysis

As shown in Table 2, the results of the univariate linear regression analysis demonstrated that age, alcohol consumption, hypertension, coronary heart disease, T2DM, stroke, glucose, TC, SBP, DBP, BMI, the taking of hypotensive drugs, and the injecting of insulin and Ln CSe were positively related to Ln LAP. In contrast, sex, race, education, and marital status were negatively correlated with Ln LAP (all *p* < 0.05). The FPRI and current smoking status were not associated with Ln LAP (*p* > 0.05).

### 3.3. Multivariate Analysis

A multivariate linear regression analysis was carried out to detect the correlation between CSe and the LAP. In model A, which had no adjustments for variables, the correlation between CSe and the LAP was positive (β = 0.76, 95% confidence interval [CI]: 0.56, 0.95; *p* < 0.001). In model B, which was adjusted for age, sex, and race, the correlation between CSe and the LAP was also positive (β = 0.66; 95% CI: 0.47, 0.84; *p* < 0.001). In model C, which was adjusted for all significant variables in the univariate analysis, the correlation between CSe and the LAP was also positive (β = 0.41; 95% CI: 0.28, 0.54; *p* < 0.001). For sensitivity analysis, we also processed CSe as a categorical variable (quartiles), and a similar trend was observed (*p* for trend < 0.001), as shown in Table 3.

### 3.4. Dose–Response Analysis

The dose–response analysis of the multivariate-adjusted linear regression was also performed (Figure 2). We found that the correlation between CSe and the LAP showed a positive non-linear manner in the log-likelihood ratio test (*p* = 0.003). In threshold effect analysis, when Ln CSe < 1.10, CSe was significantly positively related to the LAP (β = 0.45; 95% CI: 0.35, 0.55; *p* < 0.001). In contrast, when Ln CSe ≥ 1.10, CSe was not significantly positively related to the LAP (β = −0.13; 95% CI: −0.48, 0.21; *p* = 0.454) (Table 4).

### 3.5. Subgroup Analysis

A subgroup analysis was adopted to explore variable interactions (Table 5). Interaction effects were observed for age, sex, current smoking status, and stroke (all *p* < 0.05 for interactions). Conversely, no significant interactions were observed for race, education, FPRI, BMI, T2DM, glucose, coronary heart disease, hypertension, SBP, DBP, marital status, alcohol consumption, TC, the current taking of hypotensive drugs, and current injections of insulin (all *p* > 0.05 for interactions). Stronger effects were found for males (β = 0.64, 95% CI: 0.47, 0.80; *p* < 0.001) and individuals who smoked at the time of the analysis (β = 0.64, 95% CI: 0.37, 0.91; *p* < 0.001).

## 4. Discussion

As an indispensable trace element in the human body, Se plays an important role in antioxidation, anti-inflammation, anti-aging, energy metabolism regulation, etc. [13,14,15,16,17,18,19,20,21,22]. In nature, Se exists in two forms: inorganic Se and organic Se. Se is absorbed by the small intestine, and then distributed into various tissues of the body. After being absorbed by the small intestine, it is divided into various tissues of the body, which are then used to synthesize various selenoproteins that play important biological roles [14]. There are twenty-five types of human selenoproteins, all of which are very small, including five glutathione peroxidases (GPx), three thioredoxin reductases, three iodothyronine deiodinases, and others [20]. The synthesis of these selenoproteins requires the insertion of a Se-containing homolog of cysteine and 25 coding genes [13,14]. GPx1 is the most abundant selenoprotein in mammals, and it is an enzyme that is universally expressed in various cell types. Along with the consumption of reduced glutathione, GPx1 consumes reduced glutathione in order to convert lipid peroxides to their respective alcohols, and to convert H_2_O_2_ to water [17]. This physiological process is beneficial, as it alleviates oxidative damage to biomolecules such as lipids, lipoproteins, and DNA [17], in addition to maintaining membrane integrity, and reducing the related risks of various diseases [17]. Se can also intervene in energy metabolism by activating adipose tissue and regulating thyroid hormones [22].

To date, our analysis is the first to explore the correlation between CSe and the LAP. A positive non-linear correlation between CSe and the LAP was observed in our study. Moreover, the positive relationship between CSe and the LAP was more substantial in participants who were male and currently smoking. Previous literature has shown that the connection between Se levels and obesity is complex and contradictory. Previous observational research has demonstrated that the plasma Se content was negatively related to obesity among children [33]. In contrast, a case-control study of 847 adults reported that a high serum Se concentration was related to a high BMI [34]. A separate study in women revealed that hair Se levels increased in obese individuals [35]. However, a study on French adults reported that the serum Se content was not correlated with the BMI, but rather with serum cholesterol levels [36]. Furthermore, a previous NHANES study also reported that the Se dietary intake was unrelated to the BMI and WC [37]. Nevertheless, another study revealed a positive correlation between Se dietary intake and obesity in adults [38].

Not only are the results of observational studies inconsistent, but those of interventional studies are also. In animal models, the BMI of obese mice was reduced following the dietary selenomethionine intake, which facilitated the browning reaction [39]. However, a randomized prospective survey observed that the BMI was not changed, but there was a significant increase in lean muscle mass and a decrease in leptin levels after 3 months in participants taking oral 240 μg L of selenomethionine per day [40].

In the subgroup analysis, we observed that the connection between CSe and the LAP was influenced by age, sex, current smoking status, and stroke. Serum Se levels were higher in the older group [34]. It is known that differences in adipose distribution and proportions between males and females directly affect the evaluation of the LAP. In addition, lifestyle, behavior, and sex hormones differ between males and females [41,42]. The gene expression of selenoproteins differs between the sexes [43,44]. A Japanese study found a strong connection between the LAP and diabetes mellitus in both sexes [45]. The clinical features of diabetes differ between the sexes [46]. An American study reported that the whole blood Se concentration was higher in male non-smokers [47]. Our previous study found that CSe levels were negatively correlated with stroke [48], meaning that CSe levels were decreased in stroke patients. However, CSe was positively related to the LAP. This contrast amplified the relationship between CSe and the LAP, and made this relationship more significant.

This study has several strengths. First, this is the first analysis of the relationship between CSe and the LAP. Second, we found that CSe and the LAP were positively correlated in a non-linear manner. Third, the sample size of our study was relatively large.

Nevertheless, several additional limitations existed in this research. First, because our analysis was based on an observational survey, we can only draw a correlation, not a causal conclusion. Second, recall biases existed in our study due to some diseases identified based on self-reported diagnostic histories. This is despite the fact that self-reported diagnostic histories were consistent in medical records, particularly for stroke, hypertension, and diabetes mellitus [49]. Nonetheless, the individuals enrolled into our analysis were American adults. Thus, there may be inherent population bias, and further investigation is required to generalize our results to other populations.

## 5. Conclusions

In conclusion, our results indicated that CSe was positively correlated with the LAP in a non-linear manner. Future investigations are warranted to explore their relationship and better understand the mechanisms underlying this association.

## Figures and Tables

**Figure 1 nutrients-16-00933-f001:**
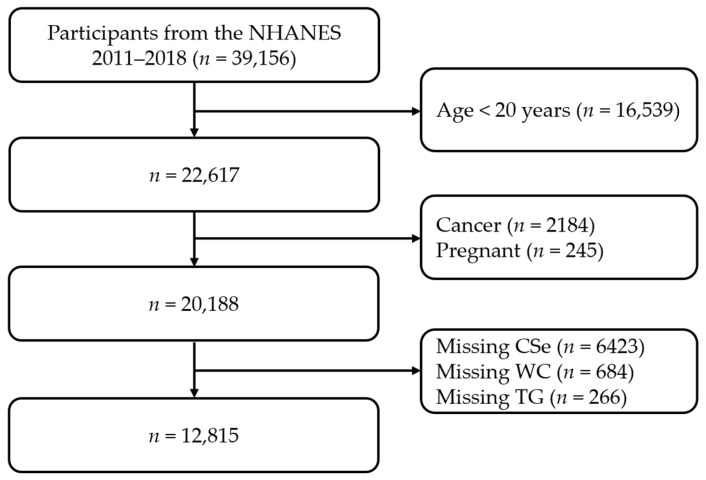
Flow diagram of screening for individuals. NHANES, National Health and Nutrition Examination Survey; CSe, circulating selenium; WC, waist circumference; TG, triglyceride.

**Figure 2 nutrients-16-00933-f002:**
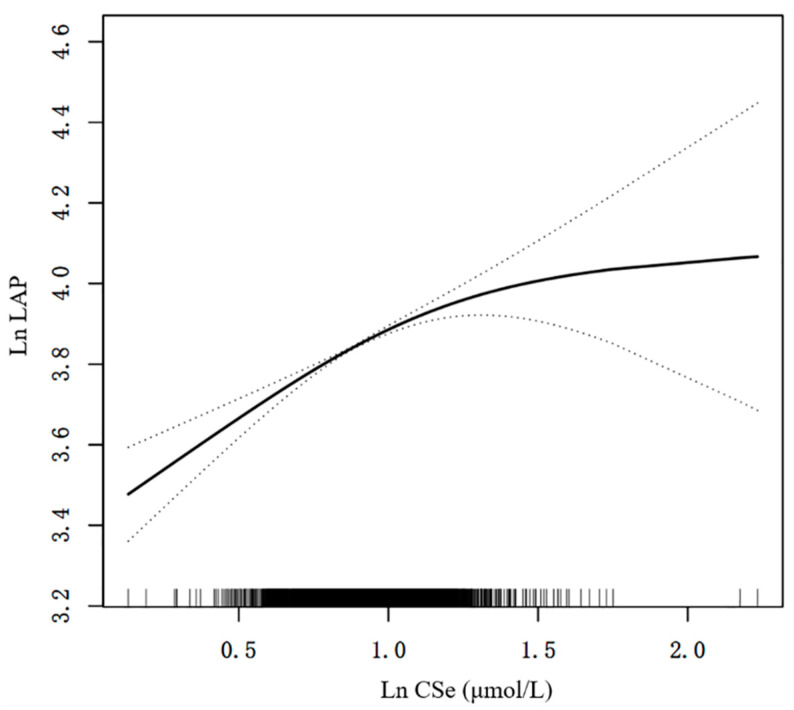
A positive non-linear manner between CSe and the LAP. A positive non-linear manner was identified after adjusting for multiple confounders in model C. The solid and dashed lines describe the β value and 95% CI, respectively. CI, confidence interval; Ln, natural logarithmic; LAP, lipid accumulation product; CSe, circulating selenium.

**Table 1 nutrients-16-00933-t001:** Baseline characteristics of individuals based on the LAP tertiles.

	Total (*n* = 12,815)	Low (*n* = 4272)	Medium (*n* = 4271)	High (*n* = 4272)	*p* Value
Age (years)					<0.001
Young	5726 (48.46)	2514 (61.80)	1672 (44.04)	1540 (39.46)	
Middle-aged	3297 (28.78)	855 (22.18)	1129 (29.98)	1313 (34.18)	
Old	3792 (22.77)	903 (16.03)	1470 (25.98)	1419 (26.36)	
Sex					<0.001
Males	6379 (49.94)	2057 (46.23)	2065 (48.77)	2257 (54.79)	
Females	6436 (50.06)	2215 (53.77)	2206 (51.23)	2015 (45.21)	
Race					<0.001
Mexican American	1754 (9.18)	390 (6.98)	594 (9.25)	770 (11.31)	
Other Hispanic	1384 (6.89)	381 (6.71)	492 (7.18)	511 (6.78)	
Non-Hispanic White	4458 (63.42)	1372 (61.53)	1431 (62.83)	1655 (65.87)	
Non-Hispanic Black	2958 (11.18)	1176 (13.81)	1051 (11.77)	731 (7.99)	
Others	2261 (9.34)	953 (10.97)	703 (8.97)	605 (8.06)	
Education					<0.001
<high school	2775 (14.02)	752 (11.45)	956 (14.32)	1067 (16.29)	
High school	2887 (23.36)	915 (21.19)	934 (21.89)	1038 (26.97)	
>high school	7144 (62.59)	2602 (67.33)	2377 (63.76)	2165 (56.71)	
Un	9 (0.03)	3 (0.02)	4 (0.03)	2 (0.03)	
FPRI					0.027
<1	2542 (13.79)	842 (14.19)	799 (12.70)	901 (14.45)	
1–3	4901 (33.83)	1540 (31.70)	1631 (34.16)	1730 (35.64)	
>3	4141 (44.86)	1486 (46.23)	1400 (45.21)	1255 (43.15)	
Un	1231 (7.52)	404 (7.88)	441 (7.93)	386 (6.76)	
Marital status					<0.001
Married	6403 (53.40)	1874 (46.85)	2221 (55.50)	2308 (57.88)	
Never	2645 (20.50)	1274 (29.67)	743 (17.40)	628 (14.38)	
Others	3761 (26.08)	1122 (23.45)	1304 (27.08)	1335 (27.73)	
Un	6 (0.02)	2 (0.03)	3 (0.02)	1 (0.01)	
Current smoking status					0.470
No	10,317 (81.19)	3407 (81.31)	3499 (81.89)	3411 (80.39)	
Yes	2491 (18.79)	861 (18.65)	770 (18.09)	860 (19.60)	
Un	7 (0.02)	4 (0.04)	2 (0.02)	1 (0.01)	
Alcohol consumption					<0.001
≤3 drinks/day	6400 (55.60)	2272 (58.65)	2144 (56.06)	1984 (52.10)	
>3 drinks/day	1855 (15.95)	617 (16.03)	555 (14.77)	683 (17.03)	
Un	4560 (28.45)	1383 (25.32)	1572 (29.17)	1605 (30.88)	
BMI (kg/m^2^)					<0.001
<25	3678 (28.32)	2686 (63.35)	751 (17.05)	241 (4.53)	
25–30	4127 (32.76)	1207 (29.05)	1792 (44.63)	1128 (25.06)	
≥30	4979 (38.74)	370 (7.60)	1716 (38.32)	2893 (70.41)	
Un	31 (0.17)	9 (0.153)	12 (0.219)	10 (0.135)	
Hypertension					<0.001
No	6286 (53.87)	2797 (70.84)	1989 (52.53)	1500 (38.25)	
Yes	6529 (46.13)	1475 (29.16)	2282 (47.47)	2772 (61.75)	
T2DM					<0.001
No	10,433 (86.60)	3966 (95.80)	3505 (88.34)	3023 (76.91)	
Yes	2382 (13.40)	306 (4.20)	762 (11.62)	1248 (23.07)	
Stroke					<0.001
No	12,388 (97.66)	4169 (98.44)	4115 (97.55)	4104 (96.99)	
Yes	420 (2.28)	100 (1.51)	155 (2.44)	165 (2.88)	
Un	7 (0.06)	3 (0.05)	1 (0.01)	3 (0.13)	
Coronary heart disease					<0.001
No	12,362 (96.97)	4180 (98.06)	4122 (97.31)	4060 (95.57)	
Yes	412 (2.82)	83 (1.83)	137 (2.47)	192 (4.16)	
Un	41 (0.20)	9 (0.11)	12 (0.22)	20 (0.28)	
Current taking of hypotensive drugs					<0.001
No	8952 (74.02)	3519 (86.16)	2896 (73.41)	2537 (62.53)	
Yes	3262 (21.18)	595 (10.15)	1170 (21.84)	1497 (31.53)	
Un	601 (4.80)	158 (3.69)	205 (4.76)	238 (5.95)	
Current injection of insulin					<0.001
No	12,336 (97.45)	4206 (98.64)	4140 (98.14)	3990 (95.59)	
Yes	474 (2.51)	65 (1.32)	129 (1.82)	280 (4.38)	
Un	5 (0.04)	1 (0.04)	2 (0.04)	2 (0.03)	
SBP (mmHg)	119.33 (110.67, 130.00)	114.00 (106.00, 124.67)	120.00 (111.33, 130.00)	123.33 (114.67, 134.00)	<0.001
DBP (mmHg)	71.33 (64.67, 78.00)	68.67 (62.67, 75.33)	72.00 (65.33, 78.00)	74.00 (67.33, 80.67)	<0.001
TC (mmol/L)	4.89 (4.22, 5.59)	4.55 (3.98, 5.20)	4.94 (4.27, 5.61)	5.17 (4.53, 5.87)	<0.001
TG (mmol/L)	1.33 (0.89, 2.00)	0.79 (0.63, 1.01)	1.30 (1.05, 1.60)	2.36 (1.83, 3.25)	<0.001
Glucose (mmol/L)	5.11 (4.72, 5.61)	4.88 (4.61, 5.27)	5.11 (4.77, 5.55)	5.38 (4.94, 6.16)	<0.001
CSe (μmol/L)	2.45 (2.27, 2.65)	2.41 (2.24, 2.60)	2.45 (2.27, 2.63)	2.49 (2.30, 2.69)	<0.001

LAP, lipid accumulation product; BMI, body mass index; SBP, systolic blood pressure; DBP, diastolic blood pressure; T2DM, type 2 diabetes mellitus; TC, total cholesterol; TG, triglycerides; CSe, circulating selenium; Un, unavailable.

**Table 2 nutrients-16-00933-t002:** Results of univariate linear regression analysis of each variable.

	β (95% CI)	*p* Value
Age (years)		
Young	Ref	
Middle-aged	0.37 (0.31, 0.43)	<0.001
Old	0.39 (0.33, 0.45)	<0.001
Sex		
Males	Ref	
Females	−0.11 (−0.16, −0.07)	<0.001
Race		
Mexican American	Ref	
Other Hispanic	−0.20 (−0.28, −0.12)	<0.001
Non-Hispanic White	−0.15 (−0.22, −0.09)	<0.001
Non-Hispanic Black	−0.44 (−0.50, −0.39)	<0.001
Others	−0.31 (−0.39, −0.24)	<0.001
Education		
<high school	Ref	
High school	−0.02 (−0.09, 0.04)	0.487
>high school	−0.17 (−0.22, −0.11)	<0.001
Un	−0.01 (−0.45, 0.42)	0.962
FPRI		
<1	Ref	
1–3	0.06 (−0.02, 0.13)	0.133
>3	−0.02 (−0.06, 0.09)	0.685
Un	−0.04 (−0.13, 0.06)	0.454
Marital status		
Married	Ref	
Never	−0.43 (−0.49, −0.36)	<0.001
Others	−0.04 (−0.09, 0.01)	0.099
Un	−0.14 (−0.55, 0.27)	0.511
Current smoking status		
No	Ref	
Yes	−0.00 (−0.06, 0.05)	0.948
Un	−0.18 (−0.95, 0.58)	0.639
Alcohol consumption		
≤3 drinks/day	Ref	
>3 drinks/day	0.05 (−0.03, 0.13)	0.200
Un	0.11 (0.06, 0.15)	<0.001
BMI (kg/m^2^)		
<25	Ref	
25–30	0.84 (0.80, 0.88)	<0.001
≥30	1.44 (1.40, 1.48)	<0.001
Un	0.89 (0.39, 1.39)	<0.001
Hypertension		
No	Ref	
Yes	0.52 (0.47, 0.57)	<0.001
T2DM		
No	Ref	
Yes	0.68 (0.61, 0.75)	<0.001
Stroke		
No	Ref	
Yes	0.24 (0.12, 0.37)	<0.001
Un	0.20 (−0.78, 1.17)	0.692
Coronary heart disease		
No	Ref	
Yes	0.37 (0.22, 0.52)	<0.001
Un	0.22 (−0.06, 0.49)	0.131
TC (mmol/L)		
<6.22	Ref	
≥6.22	0.50 (0.44, 0.56)	<0.001
Glucose (mmol/L)		
<6.1	Ref	
6.1–7.0	0.46 (0.39, 0.53)	<0.001
≥7.0	0.81 (0.72, 0.89)	<0.001
SBP (mmHg)		
<130	Ref	
≥130	0.37 (0.32, 0.41)	<0.001
Un	0.17 (0.10, 0.24)	<0.001
DBP (mmHg)		
<80	Ref	
≥80	0.37 (0.31, 0.42)	<0.001
Un	0.15 (0.08, 0.23)	<0.001
Current taking of hypotensive drugs		
No	Ref	
Yes	0.53 (0.47, 0.59)	<0.001
Un	0.31 (0.23, 0.40)	<0.001
Current injection of insulin		
No	Ref	
Yes	0.54 (0.40, 0.69)	<0.001
Un	−0.10 (−1.22, 1.01)	0.855
Ln CSe (μmol/L)	0.76 (0.56, 0.95)	<0.001

BMI, body mass index; SBP, systolic blood pressure; DBP, diastolic blood pressure; T2DM, type 2 diabetes mellitus; TC, total cholesterol; CSe, circulating selenium; CI, confidence interval; Ln, natural logarithmic; Ref, reference; Un, unavailable.

**Table 3 nutrients-16-00933-t003:** The detection of the independent relationship between Cse and the LAP using multivariate linear regression analysis.

	Model A	*p* Value	Model B	*p* Value	Model C	*p* Value
	β (95% CI)		β (95% CI)		β (95% CI)	
Ln CSe (μmol/L)	0.76 (0.56, 0.95)	<0.001	0.66 (0.47, 0.84)	<0.001	0.41 (0.28, 0.54)	<0.001
Ln Cse quartiles						
Quartile 1 (<0.81)	Ref		Ref		Ref	
Quartile 2 (0.81–0.89)	0.09 (0.02, 0.15)	0.018	0.07 (0.00, 0.14)	0.048	0.04 (−0.01, 0.10)	0.119
Quartile 3 (0.90–0.97)	0.16 (0.10, 0.21)	<0.001	0.14 (0.08, 0.20)	<0.001	0.07 (0.02, 0.12)	0.011
Quartile 4 (≥0.98)	0.29 (0.23, 0.36)	<0.001	0.26 (0.20, 0.32)	<0.001	0.16 (0.12, 0.21)	<0.001
*p* for trend		<0.001		<0.001		<0.001

Model A: no adjustment. Model B: adjusted for age, sex, and race. Model C: adjusted for age, sex, race, education, marital status, alcohol consumption, BMI, hypertension, T2DM, stroke, coronary heart disease, TC, glucose, SBP, DBP, current taking of hypotensive drugs, and current injection of insulin. CI, confidence interval; Ln, natural logarithmic; Ref, reference; LAP, lipid accumulation product; CSe, circulating selenium.

**Table 4 nutrients-16-00933-t004:** The threshold effect analysis.

Inflection Point of Ln CSe (μmol/L)	β (95% CI)	*p* Value
<1.10	0.45 (0.35, 0.55)	<0.001
≥1.10	−0.13 (−0.48, 0.21)	0.454
*p* for log-likelihood ratio test		0.003

CI, confidence interval; Ln, natural logarithmic; CSe, circulating selenium.

**Table 5 nutrients-16-00933-t005:** Interaction effects in the subgroup analysis.

	β (95% CI)	*p* Value	*p* for Interaction
Age (years)			0.032
Young	0.47 (0.33, 0.62)	<0.001	
Middle-aged	0.56 (0.26, 0.86)	0.001	
Old	0.17 (−0.02, 0.36)	0.092	
Sex			<0.001
Males	0.64 (0.47, 0.80)	<0.001	
Females	0.20 (0.06, 0.34)	0.011	
Race			0.589
Mexican American	0.30 (−0.03, 0.63)	0.094	
Other Hispanic	0.31 (−0.00, 0.63)	0.068	
Non-Hispanic White	0.40 (0.24, 0.57)	<0.001	
Non-Hispanic Black	0.41 (0.22, 0.61)	<0.001	
Other	0.57 (0.34, 0.79)	<0.001	
Education			0.981
<high school	0.42 (0.20, 0.65)	0.001	
High school	0.41 (0.18, 0.64)	0.002	
>high school	0.40 (0.25, 0.55)	<0.001	
FPRI			0.985
<1	0.41 (0.20, 0.61)	0.001	
1–3	0.38 (0.19, 0.58)	0.001	
>3	0.40 (0.20, 0.59)	0.001	
Marital status			0.210
Married	0.33 (0.17, 0.50)	0.001	
Never	0.60 (0.37, 0.83)	<0.001	
Others	0.41 (0.18, 0.64)	0.002	
Current smoking status			0.037
No	0.37 (0.25, 0.50)	<0.001	
Yes	0.64 (0.37, 0.91)	<0.001	
Alcohol consumption (drinks/day)			0.092
≤3	0.37 (0.19, 0.56)	<0.001	
>3	0.66 (0.36, 0.96)	<0.001	
BMI (kg/m^2^)			0.092
<25	0.40 (0.18, 0.62)	0.002	
25–30	0.26 (0.07, 0.44)	0.013	
≥30	0.55 (0.35, 0.76)	<0.001	
Hypertension			0.067
No	0.51 (0.35, 0.66)	<0.001	
Yes	0.31 (0.14, 0.48)	0.002	
T2DM			0.805
No	0.41 (0.28, 0.55)	<0.001	
Yes	0.38 (0.09, 0.66)	0.017	
Stroke			0.033
No	0.43 (0.30, 0.57)	<0.001	
Yes	−0.32 (−0.95, 0.31)	0.329	
Coronary heart disease			0.943
No	0.40 (0.28, 0.53)	<0.001	
Yes	0.42 (−0.01, 0.85)	0.066	
TC (mmol/L)			0.331
<6.22	0.44 (0.31, 0.57)	<0.001	
≥6.22	0.22 (−0.21, 0.65)	0.336	
Glucose (mmol/L)			0.051
<6.1	0.41 (0.28, 0.55)	<0.001	
6.1–7.0	0.07 (−0.26, 0.41)	0.678	
≥7.0	0.68 (0.31, 1.05)	0.002	
SBP (mmHg)			0.903
<130	0.41 (0.27, 0.55)	<0.001	
≥130	0.39 (0.10, 0.68)	0.014	
DBP (mmHg)			0.076
<80	0.46 (0.31, 0.61)	<0.001	
≥80	0.20 (−0.06, 0.46)	0.139	
Current taking of hypotensive drugs			0.066
No	0.47 (0.32, 0.61)	<0.001	
Yes	0.22 (−0.01, 0.45)	0.077	
Current injection of insulin			0.688
No	0.41 (0.28, 0.54)	<0.001	
Yes	0.29 (−0.31, 0.88)	0.352	

Adjusted for multiple confounders in model C, except for the subgroup variable. BMI, body mass index; SBP, systolic blood pressure; DBP, diastolic blood pressure; T2DM, type 2 diabetes mellitus; TC, total cholesterol; CI, confidence interval.

## Data Availability

The data are openly accessible via the NHANES and can be found here: https://www.cdc.gov/ (accessed on 12 March 2023).

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
