# Peer review of "The Circulating Selenium Concentration Is Positively Related to the Lipid Accumulation Product: A Population-Based Cross-Sectional Study"

_nutrients, 2024, doi:10.3390/nu16070933_

Round 1
Reviewer 1 Report
Comments and Suggestions for Authors
Comments to be shared with the authors:
The authors pose an interesting question, since biological processes that include selenium function (Se) overlap with those related to obesity, could there be a correlation between Se concentration and obesity. As a measure of obesity LAP or 'Lipid Accumulation Product" was used.
This factor is derived from waist circumference and blood triglyceride levels. However, I would not call LAP a new marker of obesity since LAP was first described in 2005 by Kahn et al. Also, rather than calling LAP a marker of Obesity, I think it would be better to consider it as a marker for Metabolic Syndrome.
The authors report that a positive correlation between CSe and LAP, and a positive non-linear trend in the fully adjusted model (model C) were observed.
In the discussion the possible reasons for the complex non-linear relationship between obesity (or LAP) and selenium concentration is explained in length.
Also the pros and cons of Se supplementation to address the negative effects of obesity are discussed. This section is comprehensive and well written.
Comments about the results presentation:
Table 1. Please indicate low/medium/ high LAP index for the Tertiles instead of just 1, 2 , 3. Table 2 and 3. Why was linear regression analysis chosen instead of a correlation analysis?
Comments on the Quality of English Language
Some corrections for the grammar or word choice are still needed. Such as when to use current vs currently.
Author Response
1.The authors pose an interesting question, since biological processes that include selenium function (Se) overlap with those related to obesity, could there be a correlation between Se concentration and obesity. As a measure of obesity LAP or "Lipid Accumulation Product" was used.
Response: Thank you very much for taking time out of your busy schedule to review our articles. Your comments and suggestions are very pertinent. We admire your dedication and academic professionalism.
According to your comments, we have made some minor revisions to this part of the statement as follows:
(LAP) is calculated from waist circumference (WC) and triglyceride (TG), and is a reliable marker of metabolic syndrome, including obesity.
This looks like a good transition and connection. We hope our modification will be satisfactory to you. If you are still not satisfied with our changes, we are still willing to revise them again.
- Table 1. Please indicate low/medium/ high LAP index for the Tertiles instead of just 1, 2 , 3.
Response: According to your suggestion, we have modified Table 1 and related text description. Replace just 1, 2, and 3 with low/medium/ high.
- Table 2 and 3. Why was linear regression analysis chosen instead of a correlation analysis?
Response: For the selection of this statistical method, we refer to several articles similar to our study and decide. The method of linear regression analysis is adopted in these articles. Compared with correlation analysis, linear regression method has some advantages, which can effectively adjust the mixing factors and make the results more accurate. In these aspects of consideration, therefore, we adopt this statistical method.
We hope our modification will be satisfactory to you. If you are still not satisfied with our changes, we are still willing to revise them again.
Reviewer 2 Report
Comments and Suggestions for Authors
1. The discussion is highly speculative and weakly based on references - „we expected to observe a negative connection between CSe and LAP, and further anticipated suggesting Se supplementation as a strategy to reduce obesity.’ - This assumption is unusual, and we know that the treatment of obesity is not related to vitamin supplementation, as in the history of chromium supplementation, modern anti-obesity drugs address the GLP system and hunger and satiety centers in the brain.
2. How can we explain the contradictory findings on the level of plasma selenium in patients with obesity, including the previous NHANES population?
3. What does „OS’ abbraviation stand?
4. „We speculate that there may also be other forms of resistance besides insulin. It has been shown that Se deficiency can cause OS in cells [45]. This suggests that OS and inflammation in obese patients are not caused by high Se levels but rather by increased Se demand. Therefore, Se may be protective against obesity. „ - What is a possible mechanism for advocating this thesis?
5. „It is known that differences in adipose distribution and proportions between males and females directly affects the evaluation of LAP. In addition, lifestyle, behavior, and sex hormones differ between males and females [52,53]. Gene expression of selenoprotein differs between the sexes [54,55]. Obesity can also affect the connection between Se and LAP as a result of adipose accumulation in patients with obesity, which can affect the assessment of the LPA. A Japanese study found a strong connection between LAP and diabetes mellitus for both sexes [56]. Clinical features of diabetes differ between the sexes [57]. A previous study reported that whole blood Se was higher in male non-smokers [58]. Our previous study found that CSe was negatively correlated with stroke [59].” - This text does not lead to any conclusion that does not have any background in the presented results, and does not apply to the conclusion of the manuscript.
6. „Furthermore, the development of obesity as a chronic disease is slow. Because the whole blood Se concentration in this study starts from around 1μmol/L, it is unlikely to fall on the U-shaped right arm. In other words, the relationship between whole blood Se level and LAP shown by our study is not due to the high level of Se, which causes the curve to fall on the right arm of the U-shaped relationship and reflects the positive correlation. „ - The authors should incorporate into the analysis the time of obesity present in the patients so that they could elaborate on the argument that the time of excess body fat influences selenium plasma levels.
7. The authors should present the data on thyroid hormones of the study population that may be affected by selenium levels of thyroperoxidase activity and thyroid hormones affecting body mass.
8. The discussion is too long and needs shortening and should be more concise with more background-orientated data and not so highly speculative.
Comments on the Quality of English Language
I found spelling mistakes
Author Response
- The discussion is highly speculative and weakly based on references - „we expected to observe a negative connection between CSe and LAP, and further anticipated suggesting Se supplementation as a strategy to reduce obesity.’ - This assumption is unusual, and we know that the treatment of obesity is not related to vitamin supplementation, as in the history of chromium supplementation, modern anti-obesity drugs address the GLP system and hunger and satiety centers in the brain.
Response: Thank you very much for your valuable time in your busy schedule to review our articles. Your comments and Suggestions are very pertinent. We admire your professionalism and academic professionalism.
As an epidemiological chapter, direct description of the results can be done. The current reference does not explain the results we find. In order to make the most effort to make a reasonable explanation, we refer to the concept of "selenium resistance", which is boldly proposed after the current literature. This concept theoretically explains the results of our discovery. Unfortunately, there is no research to prove the concept. The greatest purpose of this concept is to provide a new idea to the reader. This side of the researchers can try this idea.
In addition, there has been a testament to the fact that the supplement of selenium is effective in the treatment of obesity[PMID: 36435293], directly demonstrating that we believe that selenium is a protective effect of obesity. But statistically speaking, there is contradiction. It is not uncommon for epidemiologic research and basic research to be inconsistent, but it is not easy to find the right reasons for the explanation. The idea that we're proposing is likely to be the reason. We are conducting relevant animal experiments and clinical trials to prove the validity of this concept. We are more confident about the concept. In addition, we did not delete this paragraph because other reviewers praised the concept.
We hope our modification will be satisfactory to you. If you are still not satisfied with our changes, we are still willing to revise them again.
- How can we explain the contradictory findings on the level of plasma selenium in patients with obesity, including the previous NHANES population?
Response: Because the data we used came from NHANES. the database didn't have data on the blood selenium before 2011, so we couldn't get the NHANES data before we did it. So we took the data from the period of 2011-2018.
- What does „OS’ abbraviation stand?
Response: “OS” means oxidative stress. “OS” abbreviation was first used in section “1. Introduction”.
- „We speculate that there may also be other forms of resistance besides insulin. It has been shown that Se deficiency can cause OS in cells [45]. This suggests that OS and inflammation in obese patients are not caused by high Se levels but rather by increased Se demand. Therefore, Se may be protective against obesity. „ - What is a possible mechanism for advocating this thesis?
Response: Selenium itself is an antioxidant, with anti-oxidative stress, anti-inflammatory and other effects. Therefore, it is obviously unreasonable to say that selenium causes oxidative stress and inflammation in obese patients. Selenium plays an important role in antioxidant and anti-inflammatory activities. Obese patients have oxidative stress, inflammation, and need to consume more antioxidants, including selenium, in order to maintain balance. However, due to the problems in the utilization of selenium, a large amount of selenium cannot be converted into selenium protein. Therefore, the hemorrhagic selenium level is high.
Two key points support this view:
- The activity of selenium-containing protease in obese patients is decreased, and the activity is increased after weight loss. This phenomenon is supported by a reference cited in our manuscript discussion [PMID: 21686173]. And expressed in the manuscript, as follows:
The activity of antioxidants, including glutathione peroxidases (GPx), superoxide dismutase (SOD), and catalase (CAT), are reduced in obesity, with an increase in reactive oxidative stress (ROS) and inflammation [40]. Conversely, after weight reduction and weight loss therapy (hypocaloric dietary intervention, insulin treatment), GPx level and enzymatic activity increased.
- Does selenium have a protective effect on obesity? There is now published literature to support this view. Studies have shown that obese mice can lose weight after treatment with selenium supplementation. This also proves that selenium has a protective effect on obesity [PMID: 36435293].
Add this reference to the article and amend it as follows:
Further, the most direct evidence is that dietary supplementation of selenomethionine promotes the Browning reaction of adipose tissue in obese mice, reducing fat content and body weight [38].
We hope our modification will be satisfactory to you. If you are still not satisfied with our changes, we are still willing to revise them again.
- „It is known that differences in adipose distribution and proportions between males and females directly affects the evaluation of LAP. In addition, lifestyle, behavior, and sex hormones differ between males and females [52,53]. Gene expression of selenoprotein differs between the sexes [54,55]. Obesity can also affect the connection between Se and LAP as a result of adipose accumulation in patients with obesity, which can affect the assessment of the LPA. A Japanese study found a strong connection between LAP and diabetes mellitus for both sexes [56]. Clinical features of diabetes differ between the sexes [57]. A previous study reported that whole blood Se was higher in male non-smokers [58]. Our previous study found that CSe was negatively correlated with stroke [59].” - This text does not lead to any conclusion that does not have any background in the presented results, and does not apply to the conclusion of the manuscript.
Response: This text is a discussion of the results of the subgroup analysis. We found that there were differences in the relationship between CSe and LAP among different gender groups and whether people smoked or not. In view of these phenomena, we discuss in this paragraph. This is our purpose in writing this paragraph.
- „Furthermore, the development of obesity as a chronic disease is slow. Because the whole blood Se concentration in this study starts from around 1μmol/L, it is unlikely to fall on the U-shaped right arm. In other words, the relationship between whole blood Se level and LAP shown by our study is not due to the high level of Se, which causes the curve to fall on the right arm of the U-shaped relationship and reflects the positive correlation. „ - The authors should incorporate into the analysis the time of obesity present in the patients so that they could elaborate on the argument that the time of excess body fat influences selenium plasma levels.
Response: The NHANES survey is a cross-sectional study conducted every 2 years. The population in each survey is different, and the same population is not followed over time. So NHANES doesn't keep track of the same people's weight every year. Obesity is a slow process. So the NHANES project can't give a date for the onset of obesity. Therefore, this data is not available to us. To compensate for this, our ongoing clinical trials incorporate the weight-time factor and weigh patients regularly.
In response to this problem, we have also discussed in the paper, as follows:
As this study was a cross-sectional analysis, we cannot accurately estimate the time it takes for an individual to transition from normal weight to obesity. Therefore, a compensatory increase in the blood Se level due to the early onset of obesity cannot be ruled out. However, this scenario is most unlikely because of the two-year follow-up period for the NHANES and the large sample size.
- The authors should present the data on thyroid hormones of the study population that may be affected by selenium levels of thyroperoxidase activity and thyroid hormones affecting body mass.
Response: The NHANES project is not able to provide many blood samples for analysis and does not include thyroid hormone, thyroid peroxidase activity and thyroid hormone data. Therefore, we were unable to include these data in our analysis. But this one of your suggestions is a good reminder that we will be adding this test to our ongoing clinical trial program.
- The discussion is too long and needs shortening and should be more concise with more background-orientated data and not so highly speculative.
Response: It is a fact that blood selenium is positively correlated with LAP. As an epidemiological article, in fact, it is enough to directly explain the results. However, in order to better explain this positive correlation, we boldly proposed the concept of "selenium resistance" for this purpose. The concept is also based on current references. Unfortunately, there are no references that directly prove this concept. This is both innovative and inadequate. Due to the complexity of the relationship between blood selenium and metabolic syndrome, and a review of the current references, there is no good explanation for this problem, so we "rack our brains" to put forward this concept. The main purpose of this concept is to give readers a new angle of thinking. In the future, researchers can try to solve this problem from this direction.
Two key points support this view:
- The activity of selenium-containing protease in obese patients is decreased, and the activity is increased after weight loss. This phenomenon is supported by a reference cited in our manuscript discussion [PMID: 21686173]. And expressed in the manuscript, as follows:
The activity of antioxidants, including glutathione peroxidases (GPx), superoxide dismutase (SOD), and catalase (CAT), are reduced in obesity, with an increase in reactive oxidative stress (ROS) and inflammation [40]. Conversely, after weight reduction and weight loss therapy (hypocaloric dietary intervention, insulin treatment), GPx level and enzymatic activity increased.
- Does selenium have a protective effect on obesity? There is now published literature to support this view. Studies have shown that obese mice can lose weight after treatment with selenium supplementation. This also proves that selenium has a protective effect on obesity [PMID: 36435293].
Add this reference to the article and amend it as follows:
Further, the most direct evidence is that dietary supplementation of selenomethionine promotes the Browning reaction of adipose tissue in obese mice, reducing fat content and body weight [38].
At the same time, we are also working on clinical trials and animal studies to test this concept. In order to be able to explain this relationship as clearly as possible, we have discussed a lot of trouble. If too little is discussed, it does not seem to explain the problem clearly. At the same time, in view of other reviewers' praise for the exploration of the concept of "selenium resistance", we did not delete this paragraph.
In order to make the article more concise, we have deleted this paragraph, as follows:
Furthermore, the development of obesity as a chronic disease is slow. Because the whole blood Se concentration in this study starts from around 1μmol/L, it is unlikely to fall on the U-shaped right arm. In other words, the relationship between whole blood Se level and LAP shown by our study is not due to the high level of Se, which causes the curve to fall on the right arm of the U-shaped relationship and reflects the positive correlation.
We hope our modification will be satisfactory to you. If you are still not satisfied with our changes, we are still willing to revise them again.

Reviewer 3 Report
Comments and Suggestions for Authors
Associations between microelements and physio pathological conditions may shed light on possible correlations, and / or causations.
In this context, Zhao and the associated investigations, using a subset of the publicly available NHANES cohort (from the USA) investigate the relationship between circulating selenium concentrations (CSe) and a new metabolic parameter: the “lipid accumulation product “(LAP), finding correlation between the two parameters, and suggesting that interventions aiming at regulating CSe may represent a nutritional approach to curb the development of metabolic dysfunction.
The statistical approach is well performed, with extensive multivariate linear regression analysis performed that support the final correlations observed between LAP and CSe.
The study is worth and interesting, but I wish to present some commentaries to the authors.
One mild weakness of the study resides in the fact that the correlation between LAP and CSe may, in the end, not be causal of LAP and metabolic disease. In this context, interestingly, the authors propose the concept of “Selenium Resistance” that, in my opinion, should be better discussed in the discussion section with more ample referencing to existing work.
Section 2.2: to my understanding, authors use data from the NHANES cohort. In my opinion, section 2.2 should be removed as authors did not actually perform the measurements, and a relevant reference from the analytical work of the NHANES cohort should be placed instead.
The same criticism applied to section 2.1: this section is described almost as if the authors generated and managed the cohort. Throughout the manuscript narrative it should be made clear that this is a secondary analysis on publicly available data.
In relation to the above point, the authors may state how data and data permissions (if required) are retrieved.
Among the concluding remarks, authors refer to their own previously published work reporting on the association between CSe and stroke (ref. 59). In this context, a deeper comparison between the two studies is warranted.
Discussion section, line 3: authors state: “Se plays an important role in antioxi-
dation, anti-inflammation, anti-aging, energy metabolism regulation, etc. [13–21].” As several references are cited, I believe that a more extensive (15-20 lines?) and well thought paragraph should be placed here, that would allow the reader to learn some generalities of selenium biology without the need of going back to the cited references?
Comments on the Quality of English Language
not applicable, English language use seems fine to me
Author Response
- One mild weakness of the study resides in the fact that the correlation between LAP and CSe may, in the end, not be causal of LAP and metabolic disease. In this context, interestingly, the authors propose the concept of “Selenium Resistance” that, in my opinion, should be better discussed in the discussion section with more ample referencing to existing work.
Response:
Thank you very much for taking time out of your busy schedule to review our articles. Your comments and suggestions are very pertinent. We admire your dedication and academic professionalism.
Your question really shows us your professionalism. Because our study is a cross-sectional study, the results can only reveal correlation, not causation. This is also a common shortcoming of cross-sectional studies. For this shortcoming, we also mentioned the limitations of this study at the end of the discussion section of the manuscript.
Studies have shown that the enzyme activity in obese patients is reduced [Ref.40], which is bound to affect the conversion and biological availability of selenium. It is well known that at the beginning of a biological response, there is often a behavior called compensatory. Therefore, at the beginning of the decrease of enzyme activity in obese patients, the body will consume more selenium in order to compensate. However, as the disease progresses, obese patients will develop insulin resistance, etc., and may also develop selenium resistance. At this time, the availability of insulin and selenium proteins in each tissue will decrease, resulting in an increase in insulin and selenium in the blood. The phenomenon of elevated selenium levels in patients with high LAP also appears.
In response to your comments, we discussed the concept more fully and added the following paragraphs:
In nature, Se exists in two forms: inorganic Se and organic Se. Se is absorbed by the small intestine and distributed into various tissues of the body. After being absorbed by the small intestine, Se is divided into various tissues of the body and used to synthesize various selenoproteins to play an important biological role [14]. There are twenty-five types of human selenoproteins, all very small, including five glutathione peroxidases (GPx), three thioredoxin reductases, three iodothyronine deiodinases, and others [20]. The synthesis of these selenoproteins requires the insertion of a Se-containing homolog of cysteine and 25 coding genes [13,14]. GPx1 is the most abundant selenoprotein in mammals and is an enzyme that is universally expressed in various cell types. Along with the consumption of reduced glutathione, GPx1 consumes reduced glutathione to convert lipid peroxides to their respective alcohols, and H2O2 to water [17]. This physiological process is beneficial to alleviate oxidative damage to biomolecules such as lipids, lipoproteins, and DNA, maintain membrane integrity, and reduce the related various risk of diseases [17].
Further, the most direct evidence is that dietary supplementation of selenomethionine promotes the Browning reaction of adipose tissue in obese mice, reducing fat content and body weight [38].
Our previous study found that CSe was negatively correlated with stroke [59], which means that CSe was decreased in stroke patients. However, CSe was positively related to LAP. This contrast amplified the relationship between CSe and LAP, and make this relationship more significant.
In order to make the article more concise, we have deleted this paragraph:
Furthermore, the development of obesity as a chronic disease is slow. Because the whole blood Se concentration in this study starts from around 1μmol/L, it is unlikely to fall on the U-shaped right arm. In other words, the relationship between whole blood Se level and LAP shown by our study is not due to the high level of Se, which causes the curve to fall on the right arm of the U-shaped relationship and reflects the positive correlation.
We hope our modification will be satisfactory to you. If you are still not satisfied with our changes, we are still willing to revise them again.
- Section 2.2: to my understanding, authors use data from the NHANES cohort. In my opinion, section 2.2 should be removed as authors did not actually perform the measurements, and a relevant reference from the analytical work of the NHANES cohort should be placed instead.
The same criticism applied to section 2.1: this section is described almost as if the authors generated and managed the cohort. Throughout the manuscript narrative it should be made clear that this is a secondary analysis on publicly available data.
In relation to the above point, the authors may state how data and data permissions (if required) are retrieved.
Response: Your suggestion is very pertinent, and we have learned a valuable experience again. In addition, your suggestion will make our future work more standardized.
According to your suggestion, we have deleted the original description in Section 2.2 and part of the description in Section 2.1 and rewritten it as follows:
Our study was a secondary analysis based on the NHANES. The data analysed in our study combined four survey cycles from the NHANES (2011–2012, 2013–2014, 2015–2016, and 2017–2018).
2.2. Acquisition of variables
Age, sex, race, education, family poverty ratio of income (FPRI), marital status, and pregnant status were acquired from demographic data. BMI, WC, and blood pressure were retrieved from examination data. A history of self-reported of various diseases, medication use status, alcohol consumption, and smoking status were acquired from questionnaire data. Serum lipid, fasting plasma glucose, hemoglobin A1c (HbA1c), and whole blood Se were retrieved from laboratory data. More detailed information is available from the NHANES.
- Among the concluding remarks, authors refer to their own previously published work reporting on the association between CSe and stroke (ref. 59). In this context, a deeper comparison between the two studies is warranted.
Response: This is a study we did earlier. This study revealed that blood selenium was negatively associated with stroke in the NHANES population. That is, blood selenium levels are affected by whether or not an investigator has had a stroke. In stroke patients, blood selenium levels are decreased, which magnifies the relationship between blood selenium and LAP. Therefore, the correlation between blood selenium and LAP was more significant in stroke patients.
We added a few sentences, as follows:
Our previous study found that CSe was negatively correlated with stroke [59], which means that CSe was decreased in stroke patients. However, CSe was positively related to LAP. This contrast amplified the relationship between CSe and LAP, and make this relationship more significant.
- Discussion section, line 3: authors state: “Se plays an important role in antioxidation, anti-inflammation, anti-aging, energy metabolism regulation, etc. [13–21].” As several references are cited, I believe that a more extensive (15-20 lines?) and well thought paragraph should be placed here, that would allow the reader to learn some generalities of selenium biology without the need of going back to the cited references?
Response: The suggestion you made was a good one, and it helped us learn to write better. According to your suggestion, in order to make it more convenient for the reader to understand the biological profile of selenium without going back to the cited reference, we have properly elaborated the reference [13-21] and added a statement as follows:
In nature, Se exists in two forms: inorganic Se and organic Se. Se is absorbed by the small intestine and distributed into various tissues of the body. After being absorbed by the small intestine, Se is divided into various tissues of the body and used to synthesize various selenoproteins to play an important biological role [14]. There are twenty-five types of human selenoproteins, all very small, including five glutathione peroxidases (GPx), three thioredoxin reductases, three iodothyronine deiodinases, and others [20]. The synthesis of these selenoproteins requires the insertion of a Se-containing homolog of cysteine and 25 coding genes [13,14]. GPx1 is the most abundant selenoprotein in mammals and is an enzyme that is universally expressed in various cell types. Along with the consumption of reduced glutathione, GPx1 consumes reduced glutathione to convert lipid peroxides to their respective alcohols, and H2O2 to water [17]. This physiological process is beneficial to alleviate oxidative damage to biomolecules such as lipids, lipoproteins, and DNA, maintain membrane integrity, and reduce the related various risk of diseases [17].
We hope our modification will be satisfactory to you. If you are still not satisfied with our changes, we are still willing to revise them again.

Round 2
Reviewer 1 Report
Comments and Suggestions for Authors
Thank you for addressing my minor concerns. I am satisfied with the response and the corrections made.
Comments on the Quality of English Language
Make sure that no typos and grammar errors remain.
Author Response
- Thank you for addressing my minor concerns. I am satisfied with the response and the corrections made. Make sure that no typos and grammar errors remain.
Response: We are very grateful to you for your positive evaluation of our work once again, and for your comments and suggestions. You spent a lot of time and effort in this review process. It makes us feel very guilty. Once again, we would like to express our sincere thanks to you for your dedication and efforts to our institute.
Based on your suggestions, we have invited native English-speaking professionals to check and revise the manuscript grammar and spelling. We hope you will be satisfied with our efforts. If you are still not satisfied with our changes, we look forward to your continued guidance. Thank you again for your dedication and effort to our institute.
Reviewer 2 Report
Comments and Suggestions for Authors
1. The hypothesis and discussion on Selenium as an anti-obesity agent is unconvincing.
2. The discussion is too long and highly speculative.
3. Justifying the role of selenium requires relying on a larger number of biochemical parameters.
4. There is no data on the duration of obesity.
Comments on the Quality of English Language
Only minor spelling mistakes
Author Response
- The hypothesis and discussion on Selenium as an anti-obesity agent is unconvincing.
Response: We would like to thank you very much for your evaluation of our work again, giving us the opportunity to revise, and giving us your opinions and suggestions. Your valuable guidance is constructive to our research. You spent a lot of time and effort in this review process. It makes us feel guilty. Here, we once again express our sincere thanks to you for your dedication and efforts to our institute.
In fact, the four questions you raised reflect the same problem, which is our concept of "selenium resistance". According to your suggestions, we have made modifications to the shortcomings.
Since you think that the hypothesis and discussion of selenium as an anti-obesity agent are not convincing, we will focus on the study of the relationship between selenium and obesity, whether the two are positively correlated, negatively correlated or not, and no longer investigate whether selenium can be used as an anti-obesity agent, and demonstrate from the epidemiological perspective without investigating the underlying mechanism.
According to this theme, we delete the concept of "selenium resistance" in the discussion. The discussion has also been revised to describe only the current research status, without exploring the deep mechanism. The yellow highlights in the manuscript are changes made to the previous review. Changes in this revision are highlighted in green.
- The discussion is too long and highly speculative.
Response: According to your advice, we have deleted a lot of discussion and deleted the concept of "selenium resistance". In the discussion, we only describe the phenomenon of "positive correlation between selenium and obesity," and not the deep pathological mechanism. The deep mechanism of its deep mechanism is not an epidemiological academic chapter, which is also consistent with the conclusion of the research and research of epidemiology, and our article is not a basic study. After removing the concept of "selenium resistance", the question is clearly understood.
- Justifying the role of selenium requires relying on a larger number of biochemical parameters.
Response: According to your suggestion, our problem is mainly the concept of "selenium resistance". In order to prove this concept, we cited numerous references to make the problem as clear as possible. Now that we have removed this concept, there is no need to demonstrate this concept. At present, our work only addresses the phenomenon that selenium is linearly positively correlated with obesity. And its deep mechanism is not detailed. Therefore, it is not necessary to discuss the other biochemical parameters of selenium.
- There is no data on the duration of obesity.
Response: The issue of obesity duration is involved in the concept of "selenium resistance". We have removed this concept, and of course there are no other problems introduced by the interpretation of this concept, including the duration of obesity.
- Only minor spelling mistakes
Response: The manuscript grammar and spelling have been checked and corrected by native English speakers.
We know that you have spent a lot of time and effort in the two review processes. This makes us very uneasy. Based on your constructive comments and suggestions, we have done our best to revise the manuscript. We hope our work will be to your satisfaction. If you are still not satisfied with our changes, we look forward to your continued guidance. We look forward to hearing from you. Once again, we would like to express my sincere thanks to you for your dedication and efforts for our institute.
